# Simultaneous Prediction, Determination, and Extraction of Four Polycyclic Aromatic Hydrocarbons in the Environment Using a UCON–NaH_2_PO_4_ Aqueous Two-Phase Extraction System Combined with High-Performance Liquid Chromatography-Ultraviolet Detection

**DOI:** 10.3390/molecules27196465

**Published:** 2022-09-30

**Authors:** He Chang, Yang Lu, Yantao Sun

**Affiliations:** 1Key Laboratory of Preparation and Application of Environmental Friendly Materials, Jilin Normal University, Siping 136000, China; 2School of Chemistry, Jilin Normal University, 1301 Haifeng Street, Siping 136000, China

**Keywords:** aqueous two-phase separation, high-performance liquid chromatography, polycyclic aromatic hydrocarbons, determination

## Abstract

In this paper, a new aqueous two-phase extraction system(ATPES) consisting of UCON (poly(ethylene glycol-ran-propylene glycol) monobutyl ether)–NaH_2_PO_4_ was established, and four trace polycyclic aromatic hydrocarbons (PAHs: fluorene, anthracene, pyrene and phenanthrene) in water and soil were analyzed by high-performance liquid chromatography (HPLC)–ultraviolet detection. In the multi-factor experiment, the central composite design (CCD) was used to determine the optimum technological conditions. The final optimal conditions were as follows: the concentration of UCON was 0.45 g·mL^−1^, the concentration of NaH_2_PO_4_ was 3.5 mol·L^−1^, and the temperature was 30 °C. The recovery of the four targets was 98.91–99.84% with a relative standard deviation of 0.3–2.1%. Then UCON recycling and cyclic tests were designed in the experiment, and the results showed that the recovery of PAHs gradually increased in the three extractions because of the remaining PAHs in the salt phase of last extraction. The recovery of PAHs in the UCON recycling test was less than that in the extraction test due to the wastage of UCON. In addition, a two-phase aqueous extraction model was established based on the random forest (RF) model. The results obtained were compared with the experimental data, and the root mean square error (RMSE) was 0.0371–0.0514 and the correlation coefficient R^2^ was 96.20–98.53%, proving that the model is robust and reliable.

## 1. Introduction

Polycyclic aromatic hydrocarbons (PAHs) are a class of organic compounds containing two or more benzene or heterocyclic rings; they are widely found in the natural environment in the atmosphere, water, soil, crops, and food [1]. Their structure is stable, they are difficult to degrade, exhibit hydrophobic and oleophilic characteristics, and are readily combined with other minerals; therefore, they are also food contaminants that are often found in edible oils and maintaining safe levels thereof is essential [2]. In nature, these compounds are eliminated by biodegradation, hydrolysis, photo-cracking, and so on. Ensuring a dynamic balance of PAH contents in the environment and maintaining their safe, low concentration has been the focus of much research; however, in recent years, with the intensification of human production activities, the dynamic balance of PAHs in the environment has been destroyed, resulting in a large increase of PAHs in the environment. High levels of PAHs are often detected in a variety of soils [3]. Some experimental studies have confirmed that cigarettes also contain a variety of PAHs; inhalation causes its metabolites to participate in their absorption in bone, leading to osteoporosis, bone loss, and fracture [4]. In recent years, research on PAHs has focused on the dynamics of meat or cooking conditions because the formation and inhibition mechanisms of meat are related to the involvement of free radicals which are affected by heating temperature and time, fat precursors, antioxidants, and the presence of water in the meat model system [5].

At present, there are many methods for the enrichment, separation, prediction, adsorption, and determination of PAHs, and much research focuses on the development of a quantitative determination of PAHs in samples of different filtration materials [6,7]. The preparation of new adsorbents with good adsorption efficiency and capacity can provide a better prediction for environmental pollution and bioaccessibility evaluation [8]. There were also chemical experiments in solvent-based methods that applied pre-treatment extraction techniques such as Soxhlet extraction [9,10], ultrasonic-assisted solvent extraction [11,12], microwave-assisted solvent extraction [13,14], accelerated solvent extraction [15,16], and supercritical fluid extraction [17] for cleaning and preconcentration procedures. The extract was then separated by solid phase extraction (SPE) or liquid chromatography to remove the interfering compounds in the solution. Gas chromatography–mass spectrometry (GC-MS) [18,19] and liquid chromatography–fluorescence (LC-FLD) [20,21] are also have many applications. In addition, thermal desorption methods are often used for determining granular PAHs and their derivatives [22,23]. However, there are some inevitable problems in the practical application of the above methods, such as low desorption efficiency for compounds with high boiling points (above 300 °C), complex preparation of some adsorbents, long operation time, and low enrichment coefficient; thus, they have not been widely used. In recent years, a two-phase system has had a good application prospect in the separation and purification of biological substances and has been widely used because of its advantages of environmental friendliness, low toxicity, and good biological compatibility. This method has been successfully applied in the purification of glycyrrhizin (GA) and glycyrrhizin (LQ) from glycyrrhizin [24], extraction of phenolic compounds [25], separation of polysaccharide mixtures (dextran and dextran sulphate) [26], separation and purification of papain crude extract from papaya milk [27], protein separation [28], and penicillin separation [29]. Compared with traditional organic solvent extraction and solid phase extraction, aqueous two-phase extraction has the advantages of environmental friendliness, continuous operation, and easy amplification, which also promotes the rapid development of green separation technology [30]. In previous works, the two-phase extraction technology was used to separate and enrich trace chloramphenicol in prawn [31]; trace amounts of sulfadiazine and sulfamethazine in food and environment were determined and the solanine and solanum nigrum polysaccharide in unripe fruit of Solanum nigrum were isolated and analyzed [32]; trace fluoroquinolones in environmental samples were determined [33]; and the determination and correlation of equilibrium conditions of aqueous two-phase systems composed of different ionic liquid polymers and different salt solutions were explored [34]. The experimental results showed that the aqueous two-phase extraction system(ATPES) is an environmentally friendly technology.

With the rapid development of computer science and the rapid improvement of computer software and hardware performance, artificial intelligence algorithms based on machine learning have been widely applied in many fields. Today, machine learning algorithms can quickly find correlations between variables in large volumes of data, helping researchers discover more specific patterns. As a new method, it has been applied in chemistry [35]. The advantage of machine learning in chemical research lies in its high efficiency. Machine learning algorithms can reduce the difficulty of establishing complex data models, process unstructured data, efficiently use and process large quantities of data generated by experiments, and also effectively use historical data [36]. It is suitable for analyzing complex process problems and some abstract problems which are not easy to model mathematically. In recent years, various algorithm tools have been continually improved, and increasingly more researchers have applied machine learning to chemical research, opening a new chapter in chemical research. Random forest (RF) is an integrated learning model based on multiple regression trees, which has higher prediction accuracy and better generalization performance than single-tree regression. RF further introduces random attribute selection in the training process of decision trees on the basis of bagging constructed by a decision tree-based learner [37]. During tree training, the method of bootstrap sample was adopted to randomly select samples and keep the samples independent. In the internal node-splitting process of the tree, instead of all features, a random sample of features is included in the splitting candidate. In this way, the correlation between the base models is reduced, which in the formula for the variance, continues to lead to a reduction in the overall variance. In addition, the stochastic forest model can efficiently process the data set, and the prediction accuracy is high. The data pre-processing is simple without variable screening, and the sub-models are independent from each other and will not be affected by outliers and noises. Ranking the importance of explanatory variables, it can process continuous variables and classified variables at the same time and will not be influenced by outliers in the training set, which improves the robustness of the model. Here, a UCON (polyethylene glycol-propylene glycol-butyl ether)–NaH_2_PO_4_ ATPES was established, and it was applied to high-performance liquid chromatography (HPLC) for the determination of four representative PAHs (pyrene, anthracene, phenanthrene, and fluorene). The experimental data were used as the data set, and the RF method was used to establish the model. The predicted values were compared with the real values and generalized to the separation prediction of different substances of the same type.

## 2. Experimental Work

### 2.1. Materials

UCON was obtained from Sigma America Reagent Company (average Mn ~ 3900). Organic salts (NaH_2_PO_4_, K_2_HPO_4_, (NH_4_)_2_SO_4_, K_3_PO_4_, KOH, K_2_CO_3_, KNaC_4_H_4_O_6_·4H_2_O, and Na_2_SO_4_) were analytical grade reagents (GR, the lowest mass fraction is 99%), which were purchased from the Sinopharm Chemical Reagent Co., Ltd. (Shanghai, China). The four PAHs targets (benzopyrene, anthracene, phenanthrene, and fluorene) were purchased from Aladdin Reagent (Shanghai, China) and MacLin Biochemical Co., Ltd. (Shanghai, China). All reagents were used without further purification and the water used in experiments was double distilled. Ethanol was an analytical grade reagent (GR, with a minimum ethanol mass fraction of 99.8%), and acetonitrile was an HPLC grade reagent.

### 2.2. Preparation of Stock Solution

The amount of 20 mg of each of the four PAHs were put into a beaker, acetonitrile was added, and the mixture was stirred until the target substance was completely dissolved. A standard mixture with a concentration of 200 ug·mL^−1^ was prepared to a constant volume in a 100-mL volumetric flask. Specimens were stored in a dark place and we remixed the standard solution every two months.

### 2.3. Preparation of Real Samples

Water samples were taken from Xiasantai River and Tashan Reservoir in Siping City, Jilin Province (China) and stored in 2.5 × 103 mL amber glass bottles. One day later, all the samples were centrifuged at 2000 rpm for 10 min, and then the supernatant was collected. The supernatant was filtered through a 0.45-mm filter, and the PAH working solution was added. Finally, the specimens were stored at 4 °C in a refrigerator for the further use.

Soil samples were prepared by adding pyrene, anthracene, phenanthrene, and fluorene to soil. The topsoil and subsoil were collected from vegetable fields in suburban areas of Siping City, which had been dosed with the same fertilizer for two years at a constant rate. The standard solution of the PAH mixture sample was added to the soil sample and the soil samples were then mixed and placed into a culture dish. One week later, 5 g of the soil specimens was placed into 20-mL colorimetric tubes and 10 mL of buffer (0.05·mol L^−1^ EDTA + 0.06·mol·L^−1^ Na_2_HPO_4_ + 0.08 mol·L^−1^ C_6_H_8_O_7_) was added to each tube. Each sample was oscillated for 10 min and subsequently centrifuged for 10 min at 1000 rpm. The supernatant was stored at 4 °C after filtration through a 0.45 μm filter.

### 2.4. Apparatus and Procedure

HPLC (Agilent 1100, Agilent Technologies Inc., Palo Alto, California, USA) equipped with ultraviolet–visible (UV) detector was used. An analytical balance (BS124S, Beijing Sartorius Instrument Co., Ltd., Beijing, China) with an uncertainty of ±1.0 × 10^−7^ kg was used for each weighing. The temperature of the system was controlled by a constant-temperature water bath provided by Zhanyun Biological Technology Instrument Co., Ltd. (Shanghai, China). A control machine and data processer ran Agilent ChemStation software. A Cence-H1650 centrifuge from Xiangtan Xiangyi Instrument Co., Ltd. (Xiangtan, China) was used for centrifugation.

UCON and salt in the original solution were put into a container, and real samples containing four types of target PAHs were added, and then water was added to a volume of 10 mL. The concentration of UCON was 0.09 g·mL^−1^, and the concentration of Na_2_HPO_4_ was 4 mol·L^−1^. The mixed solution was stirred continuously for 20 min and then put into a constant temperature water bath.

After the formation of the two-phase system, the two phases were separated, 1 mL of the upper phase was added with acetonitrile to 5 mL, and the concentration of the four target PAHs in the upper phase was determined by HPLC. The HPLC–UV determination method was described as follows: in chromatographic separation, an Agilent TC-C18 (2) was used to explore the reversed phase column (No. USEGK03761). The contents of four PAHs were determined at a column temperature of 30 °C with the injection volume of 10 μL and the mobile phase ratio of A:B = 65:35 (where A and B represent acetonitrile and water, respectively). The flow rate was 1.0 mL·min^−1^. The liquid was detected by ultraviolet detector at 250 nm. The operating process is shown in Figure 1.

### 2.5. Determination of the Partition Parameters of PAHs

The distribution and enrichment efficiency of four PAHs targets were characterized by enrichment coefficient and extraction efficiency. The enrichment factor (F) was determined by the ratio of the concentration of the four PAHs targets in the top phase to the concentration of the initial system.
(1)$$F=\frac{C_{t}}{C_{s}}$$
where the C_t_ is the concentration of four target PAHs in the top phase and C_s_ represents the concentration of four target PAHs in the initial system before two-phase separation. Extraction efficiency (E) is calculated thus:(2)$$E=\frac{C_{t}\times V_{t}}{M_{s}}$$
where C_t_ is the concentration of four target PAHs in the top phase, V_t_ denotes the column of top phase, and M_s_ is the total mass of four target PAHs added in the initial system. The phase ratio (R%) was defined as a ratio of the volume of top phase to the volume of bottom phase.
(3)$$R\%=\frac{V_{t}}{V_{b}}\times 100$$
where V_t_ and V_b_ refer to the volume of the top phase and bottom phase, respectively.

The achievements of the proposed ANN model and response surface methodology (RSM) were statistically measured by mean relative percent deviation (MRPD) and root mean square error (RMSE), defined as follows:(4)$$\mathrm{MRPD}=\frac{100}{n}\sum_{i=1}^{n}\left|\frac{y_{c,i}-y_{e,i}}{y_{e,i}}\right|$$
(5)$$\mathrm{RMSE}=\sqrt{\frac{\sum_{i=1}^{n}{\left(y_{c,i}-y_{e,i}\right)}^{2}}{n}}$$
where y represents the response and the subscripts c and e represent the computational data and experiment data, respectively. The subscript i represents the number of items in the dataset, and n is the total size of the dataset.

### 2.6. Statistical Analysis

Statistical analysis allows us to gather hidden information and models for estimation and prediction. RSM is a powerful method with which to evaluate and optimize the interactive effects of the prevailing phenomena. Here, the single factor test was used to set the range of optimized conditions for the extraction efficiency of four targets in the present study. The optimized conditions were determined in the multifactor experiment. RSM was used to determine the optimal conditions for the extraction and determination of trace PAHs in an aqueous two-phase extraction system (ATPES). RSM includes factorial design and regression analysis. In this experiment, the factorial design of RSM adopted a central composite design (CCD) with three variables at three levels. The independent variables were temperature (X_1_), the concentration of UCON (X_2_), and the concentration of NaH_2_PO_4_ (X_3_). Design-Expert 8.0 was used to apply CCD. These three factors and their levels and ranges are listed in Table 1.

The three levels of X_1_ are −1 (25 °C), 0 (30 °C), and 1 (35 °C); the three levels of X_2_ are −1 (0.35 g·mL^−1^), 0 (0.45 g·mL^−1^), and 1 (0.55 g·mL^−1^); and the three levels of X_3_ are −1 (2.5 mol·L^−1^), 0 (3.5 mol·L^−1^), and 1 (4.5 mol·L^−1^).

The experimental data were analyzed by a response surface regression program, and the second-order polynomial equation was used as the analytical model:(6)$$Y=\beta_{0}+\sum_{i}\beta_{i}X_{i}+\sum_{i}\beta_{\mathrm{ii}}X_{i}^{2}+\sum_{\mathrm{ij}}\beta_{\mathrm{ij}}X_{i}X_{j}\;$$
where X_i_ and X_j_ represent the factors, Y is the response, and $\beta_{0}$, $\beta_{i}$, $\beta_{\mathrm{ii}}$, and $\beta_{\mathrm{ij}}$ are the regression coefficients for the intercept, linear, quadratic, and interaction coefficients, respectively. The statistical significance of the model was evaluated by an F-test.

An RF was used to build a model to simulate the extraction and separation process. The system temperature (X_1_), the concentration of UCON (X_2_), and the concentration of NaH_2_PO_4_ (X_3_) were used as input variables, and the extraction efficiencies (Y) of four PAHs (PHE, ANT, FLU, and PYR) were used as output variables. The model was optimized by adjusting various parameters such as forest size, leaf number, and tree roots. Meanwhile, to calculate the correlation coefficient of training data, the relative importance of variables was calculated and sorted, and the data results were visualized.

## 3. Results and Discussion

### 3.1. Single Factor Experiment Results and Discussion

#### 3.1.1. Phase Behavioral Study of the UCON–Organic Salt ATPES

The formation of the double water phase is closely related to the salting-out ability of the salt. The salt composition in the mixed solution directly determines the difficulty of dividing the double water phase into two phases. The difference of salting-out ability is related to the valence of electrolyte and ion radius. In this paper, the applicability of organic salts (NaKC_4_H_4_O_6_·4H_2_O) and inorganic salts (K_2_HPO_4_, (NH_4_)_2_SO_4_, K_3_PO_4_, KOH, K_2_CO_3_, Na_2_SO_4_, and NaH_2_PO_4_) to form the ATPES with UCON is discussed. The results show that an ATPES can be formed by mixing an appropriate amount of salt with the UCON solution. We judged the phase-forming ability of organic salts according to the value of sample recovery. In our opinion, the phase-forming ability of organic salts has the following two characteristics: the first is that the order of phase forming ability of organic salts with the same cation (K^+^) is hydrogen phosphate > tartrate > carbonate, and the order of phase-forming ability of organic salts with the same cation (Na^+^) is hydrogen phosphate > sulphate > tartrate. The second problem is that the order of phase formation ability of organic salts with the same anionic (PO_4_^2−^) structure is sodium salt and potassium salt, and the order of phase formation ability of organic salts with the same anionic (SO_4_^2−^) structure is sodium salt and ammonia salt. Finally, three salts (NaH_2_PO_4_, NaKC_4_H_4_O_6_·4H_2_O, and K_2_HPO_4_) were selected to study the extraction efficiency of UCON for four PAHs. To discuss the extraction efficiency and enrichment factor of the ATPES containing three salts, PAHs were added to these systems. The changes in extraction efficiency and enrichment factors of various ATPESs are shown in Figure 2.

The extraction efficiencies of four PAHs by the ATPES containing UCON and NaH_2_PO_4_, NaKC_4_H_4_O_6_·4H_2_O, and K_2_HPO_4_ were assessed. The UCON–NaH_2_PO_4_ ATPES is shown to be superior to other salts in terms of its extraction efficiency and enrichment factor. The reasons for this are as follows: among several salt solutions, NaH_2_PO_4_ aqueous solution is acidic (pH value of 3.0 mol·L^−1^ salt solution is 3.6), NaKC_4_H_4_O_6_·4H_2_O and K_2_HPO_4_ aqueous solutions are slightly alkaline, and PAH mixed aqueous solution is weakly acidic (pH 5.83). ATPES composed of UCON–NaH_2_PO_4_ are more suitable for the extraction of PAHs because they will form PAHs salts under acidic conditions. In addition, H_2_PO_4_^−^ has a stronger association with water than HPO_4_^−^, so NaH_2_PO_4_ was selected as the phase-forming salt in the ATPES.

#### 3.1.2. Influence of the Concentration of NaH_2_PO_4_ on Distribution of PAHs

NaH_2_PO_4_ is added to the mixed solution containing polymer. Due to the strong association ability of H_2_PO_4_^−^ with water, many water molecules combined with H_2_PO_4_^−^, so that UCON gradually precipitated water phase and separated from water to form an ATPES. Therefore, the concentration of NaH_2_PO_4_ affected the formation of the ATPES. Meanwhile, the addition of NaH_2_PO_4_ reduced the activity of water molecules in the solution and increased hydrophobicity, which was conducive to extraction. We found the same trend arising in the polymer concentration and the system temperature under certain conditions, so we assayed the concentrations (0.5 mol·L^−1^, 1.5 mol·L^−1^, 2.5 mol·L^−1^, 3.5 mol·L^−1^, 4.5 mol·L^−1^, and 5.5 mol·L^−1^) of the double water phase-split phase and the change in recovery there as follows: first, 0.5 mol·L^−1^ salt solution was not separated (the upper phase height was very small (less than 0.5 mL) and disappeared after vibration); secondly, the upper phase volume decreased with the increase of the salt concentrations in other groups; and thirdly, when the salt concentration increased from 1.5 mol·L^−1^ to 4.5 mol·L^−1^, the aqueous phase was more stratified, and with the addition of polymers, more PAHs accumulated in the upper phase, the aqueous phase was more stratified, and extraction efficiency and enrichment factors increased rapidly due to the salting-out effect. Figure 3a,b shows the effect of the concentration of salt on the extraction efficiency and enrichment factor.

#### 3.1.3. Influence of the UCON Concentration on Distribution

The optimal concentration range of UCON was determined. Under the condition of constant NaH_2_PO_4_ concentration and temperature, the influence of the UCON concentration on extraction efficiency and enrichment factor was studied. Five concentration gradients were designed in the experiment (0.15 g·mL^−1^, 0.25 g·mL^−1^, 0.35 g·mL^−1^, 0.45 g·mL^−1^, and 0.55 g·mL^−1^). In the experiment, it was found that the upper phase at 0.15 g·mL^−1^ concentration was less than 1 mL in volume, so only the other four concentration gradients were investigated. By calculating the recovery rate, analysis of the recovery rate of the target substance showed a trend of first increasing, then decreasing, and the best effect was achieved when the concentration was 0.35 g·mL^−1^. The results showed that the extraction efficiency increased with increasing UCON concentrations, indicating that several target substances were more easily separated to the top phase with the increase of the UCON concentration. However, the enrichment coefficient fell because the volume of the top phase increased with the increase of polymer concentrations. Although the polymer carried more PAHs into the top phase, PAHs exist at lower concentrations which led to the reduction of the enrichment coefficient. Therefore, in the multi-factor experiment, the dosage range of UCON should be set to balance extraction efficiency and the enrichment factor (Figure 3c,d).

#### 3.1.4. Influence of System Temperature on Distribution

In addition to the concentration of polymer and salt solution, temperature is another important factor affecting the phase-formation behavior of two-phase aqueous extraction [32]. The influence of temperature within the range from 25 °C to 75 °C on extraction efficiency and enrichment factor was studied (Figure 3e,f). During the experiment, it was found that the higher the temperature, the clearer the upper phase; the partitioning efficiency of the three objects decreased with the increase of the system temperature, but the decrease was small. In theory, extraction experiments can be conducted at higher temperatures to obtain higher extraction efficiency and enrichment factor. We speculated that in this experiment, we would find that high temperature inhibits the segmentation of objects.

### 3.2. The Multi-Factor Experiment

#### 3.2.1. Design of Orthogonal Table

In Design-Expert 8.0, an RSM was used to design multi-factor experiments based on the CCD method. The optimum conditions for the extraction and determination of trace PAHs in the ATPES were determined by a single-factor test. The experimental data of the partition efficiency of the three materials were analyzed by multiple regression analysis. The coefficient of the model was significantly evaluated, and the non-significant factors were ignored to conduct multi-factor and multi-level tests to determine the optimal test conditions. Three factors were set in this experiment, and the number of levels of each factor was also set to three for the factorial design. The factors were temperature (X_1_), the concentration of UCON (X_2_), and the concentration of NaH_2_PO_4_ (X_3_). The design and results of the multi-factor experiment are listed in Table 2.

#### 3.2.2. Regression Analysis

Analysis of variance (ANOVA) was used to evaluate the statistical significance of the model. Statistical analysis showed that the extraction efficiency and enrichment factor model were significant (*p* < 0.05), but insufficient fitting data were not significant (*p* > 0.05). The lack of fit was greater than 0.1, indicating that the lack of fit was insignificant; therefore, these three models are suitable for predicting and evaluating the partition efficiency of the three materials. The experimental results were fitted using Equation (6). The R^2^ values of extraction efficiency of FLU, ANT, PYR, and PHE were 0.9840, 0.9833, 0.9837, and 0.9836, respectively. The R^2^ values of enrichment coefficient of FLU, ANT, PYR, and PHE were 0.9928, 0.9842, 0.9843, and 0.9800, respectively. The R^2^ values of their extraction efficiency and enrichment factor models were all above 0.9800, indicating that the model can represent the experimental results. It is believed that the model can be used to fit the extraction efficiency and enrichment coefficient of FLU, ANT, PYR, and PHE in PAHs, and that the results were reliable. The residual normal diagram fitting the FLU, ANT, PYR, and PHE extraction efficiency and enrichment factor models is shown in Figure 4. Most of the points in all the graphs were near or overlapped the diagonal; there were some points outside the straight line, indicating that there was a small deviation between the predicted and experimental values. Fisher’s F-test was used in ANOVA to conduct statistical analysis on the partition efficiency of the three materials (Appendix A). These results show that the model was valid in all eight groups. FLU, ANT, PYR, and PHE were studied by the HPLC–UV method, and the corresponding HPLC chromatogram is shown in Figure 5. Finally, the extraction efficiency and enrichment coefficient of FLU, ANT, PYR, and PHE were analyzed using the quadratic equations in Appendix A.

#### 3.2.3. Response Surface Plot

The 3D response surface plots, the two responses (F and E%), and the variable interaction relationship between visual experiment levels can be directly observed at the three response and the experimental level variables. The relationship between the variables facilitates observation of the interaction between two factors, and at the same time provides a direct observation of interactions between the two test-variable methods. The interaction between two variables and its optimal range can be well observed by the response surface diagram. The influences of the three factors on the partition efficiency of the four substances are shown in Figure 6. The optimal range of variables was determined by observing the response surface plots. The optimal experimental conditions were described as follows: the concentration of NaH_2_PO_4_ was 3.5 mol·L^−1^, the temperature was 30 °C, the concentration of polymer was 0.45 g·mL^−1^, and the extraction efficiencies of FLU, ANT, PHE, and PYR were 0.9891, 0.9972, 0.9917, and 0.9984, respectively. The enrichment coefficients of FLU, ANT, PHE, and PYR were 5.02, 6.23, 5.59, and 4.6, respectively.

### 3.3. The RF Model and Prediction

The experimental data were taken as a data set, and there were 181 datapoints overall. All the data were divided into training sets and test sets, in the ratio of 163:18. During training, the model did not see the data in the validation and test sets. By using invisible data, overfitting or underfitting of the model can be avoided to a large extent, and a generalized model is obtained. In this model, the number of leaves (leaf) is set to 5, the size of forest (ntrees) is set to 800, and the root of trees (nfoot) is set to 1. To prevent overtraining, cross-validation technology is used to obtain the best weight set. We determined the relationship between variable weight and grade, calculated the correlation coefficient, evaluated the error of training set and test set, respectively, and visualized the results; regarding the test dataset, the regression R-values of the proposed RF model are shown in Figure 7. The sorting results of input parameter importance are illustrated in Figure 8.

#### Generalization Ability

The proposed RF model and RSM were evaluated with completely unseen data to test their generalization abilities. The unseen data were taken from the literature such that the partition efficiencies of some objects in various ATPESs matched previous analyses. The type of ATPESs, the extraction object, the ranges of variables, the number of data, and the MRPD values for the RF model and RSM model are listed in Table 3. It was found that the RF model had better generalization ability than the RSM model as evinced by the MRPD values. It is worth mentioning that the RF model showed significant generalization ability for predicting the partition efficiency of different object in various types of ATPESs in wide ranges of operational variables.

### 3.4. Secondary Separation and Cyclic Test

To obtain greater recovery, a secondary separation experiment was designed. Based on the previous experimental operation, the heating and phase separation operations were conducted. The concentration of the salt solution (NaH_2_PO_4_) was set to 4 mol·L^−1^, and 8 mol·L^−1^ of salt solution, 1 mL standard solution, and 6 mL polymer with water were added to the colorimetric tube in a constant volume of 20 mL. The phase separation results were recorded after standing for 25 min. Taking 2 mL of upper phase, 1 mL of water and 1 mL of acetonitrile were added and mixed evenly, with a total of 4 mL of mixed solution. Then, 1 mL of this was taken directly to measure the liquid phase and the remaining 3 mL was poured into the colorimetric tube and put into the water bath at 70 °C for secondary heating for 15 min. Afterwards, the phase separation results were recorded and the upper phase test liquid phase was taken. In the process of secondary heating, experiments on switch cover heating, process heating, and direct heating were conducted. The experimental process is shown in Figure 9. By calculating the recovery rate of standard addition and comparing the experimental data, the recovery rate was found to be higher with open cover heating.

The phase containing UCON was transferred to a new centrifuge tube after the first extraction. UCON was diluted by adding distilled water, and the centrifuge tube was placed into thermostat water bath. The new two phases appeared when the system temperature was changed. The recovery efficiency was determined when the volume of the added water changed from 0.5 mL to 3.0 mL in increments of 0.5 mL. It was found that the recovery efficiency reached its maximum value at 2.0 mL. In the cyclic test, the recycled UCON was used in the next extraction experiment. The three times cyclic experiment results are listed in Table 4. The UCON losses were approximately 0.4 mL in each cyclic test, and this was mainly caused by the wastage in transfer and detection process. To form the next ATPS, the extra EOPO made up for any deficiency. The recovery of PAHs gradually increased in the three extractions because of the remaining PAHs in the salt phase of last extraction. The recovery of PAHs in UCON recycling test was less than that in the extraction test due to the wastage of UCON.

### 3.5. The Validity of the Method

The UCON–NaH_2_PO_4_ ATPES combined with HPLC was applied to the analysis of the real samples spiked with the analytes at six different concentration levels. The real samples were extracted and injected in triplicate to obtain the calibration curves (Table 5). This UCON–NaH_2_PO_4_ ATPES was applied to separate and enrich PHE, ANT, FLU, and PYR in real samples (water and soil) under the aforementioned optimal conditions. The extraction efficiencies of PHE, ANT, FLU, and PYR are listed in Appendix A. The recoveries of PHE, ANT, FLU, and PYR in spiked real samples were 96.71–99.84% when the concentration of the spiked PHE, ANT, FLU, and PYR was 0–1000 ng·mL^−1^. The accuracy and the precision of this method were evaluated by performing five replicates of the spiked samples in the same day and during seven different days. The intra- and inter-day precisions were expressed as relative standard deviations ranging from 0.23 to 2.33% and 0.11–2.45%. All this illustrated that this technique has a satisfactory recovery and reproducibility for the determination of the PHE, ANT, FLU, and PYR in real samples. Thus, this method can be used for the synchronous quantitative analysis of residual trace PHE, ANT, FLU, and PYR in the environment water and soil.

#### 3.5.1. Comparisons of Different Methods for the Determination of PHE, ANT, FLU, and PYR in Real Samples

The results are summarized in Table 6. After comparing the proposed method with those reported in the literature, the recovery efficiency, reproducibility, accuracy, and precision of this method were found to be better than those of some reported methods; moreover, it is time-saving and convenient, which is another advantage of this method besides its inherent eco-friendly recycling.

#### 3.5.2. Application of UCON–NaH_2_PO_4_ ATPES in the Separation of Trace Sulfadiazine and Sulfadimethazine from Food and Environment

Lastly, a UCON–NaH_2_PO_4_ ATPES coupled with HPLC was used to analyze trace synchronously sulfadiazine (SDZ) and sulfamethazine (SMT) in animal by-products (i.e., egg and milk) and environmental water sample, and it was found that the recovery of SDZ and SMT was 96.09–99.40% with RSD of 0.41–3.09% (Appendix A). All this illustrated that this technique has a satisfactory recovery and reproducibility for the determination of the SDZ and SMT in real samples. This proves that this method can be applied to the synchronous quantitative analysis of residual trace SDZ and SMT in the environment water and food, and it gives us enlightenment for the future research on the possibility of applying of UCON–NaH_2_PO_4_ ATPES to extract other trace sulfonamides.

## 4. Conclusions

A UCON–NaH_2_PO_4_ ATPES has been successfully applied to trace PAHs in the environment; PHE, ANT, FLU, and PYR were isolated and purified. RSM was used to determine the optimal extraction efficiency (E%) and enrichment factor (F) of these four targets through multi-factor experiments. Under optimized conditions, trace amounts of PHE, ANT, FLU, and PYR in specimens were quantitatively determined by HPLC. The optimal extraction process was described as follows: the concentration of NaH_2_PO_4_ was 3.5 mol·L^−1^ at a temperature of 30 °C, the concentration of polymer was 0.45 g·mL^−1^, and the extraction efficiencies of FLU, ANT, PHE, and PYR were 0.9891, 0.9972, 0.9917, and 0.9984, respectively. The enrichment coefficients of FLU, ANT, PHE, and PYR were 5.02, 6.23, 5.59, and 4.6 respectively. In addition, UCON recycling and cyclic tests were also designed in the experiment. The work also provides insights into whether this simultaneous extraction and separation method can be applied to other food or environmental samples.

## Figures and Tables

**Figure 1 molecules-27-06465-f001:**
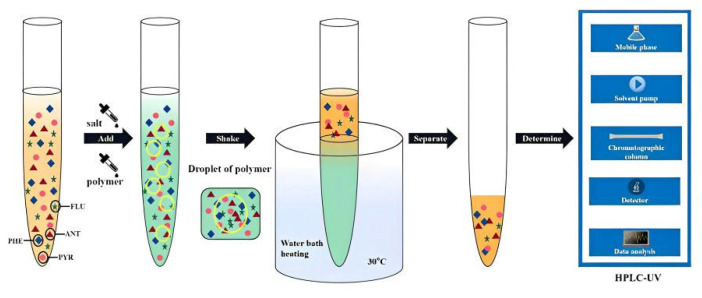
Schematic of the separation and determination of trace four PAHs.

**Figure 2 molecules-27-06465-f002:**
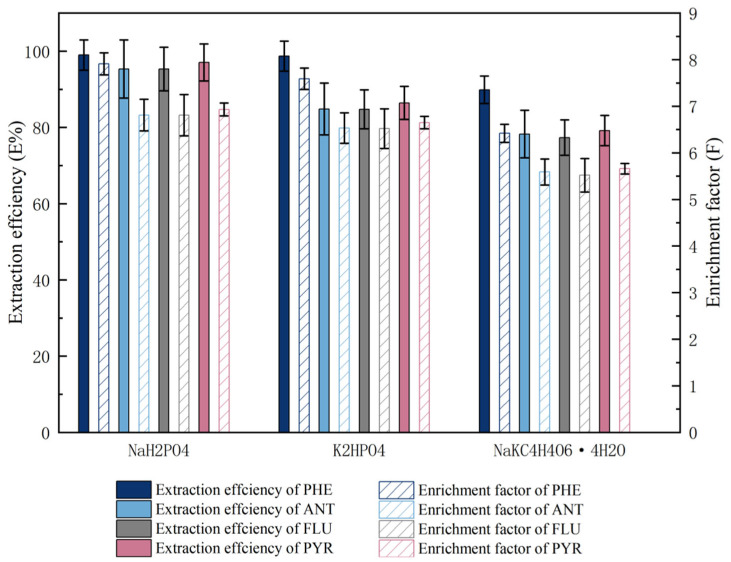
The extraction efficiency and enrichment factor of PHE, ANT, FLU and PYR in UCON–salt (NaH_2_PO_4_, NaKC_4_H_4_O_6_·4H_2_O, and K_2_HPO_4_) ATPESs.

**Figure 3 molecules-27-06465-f003:**
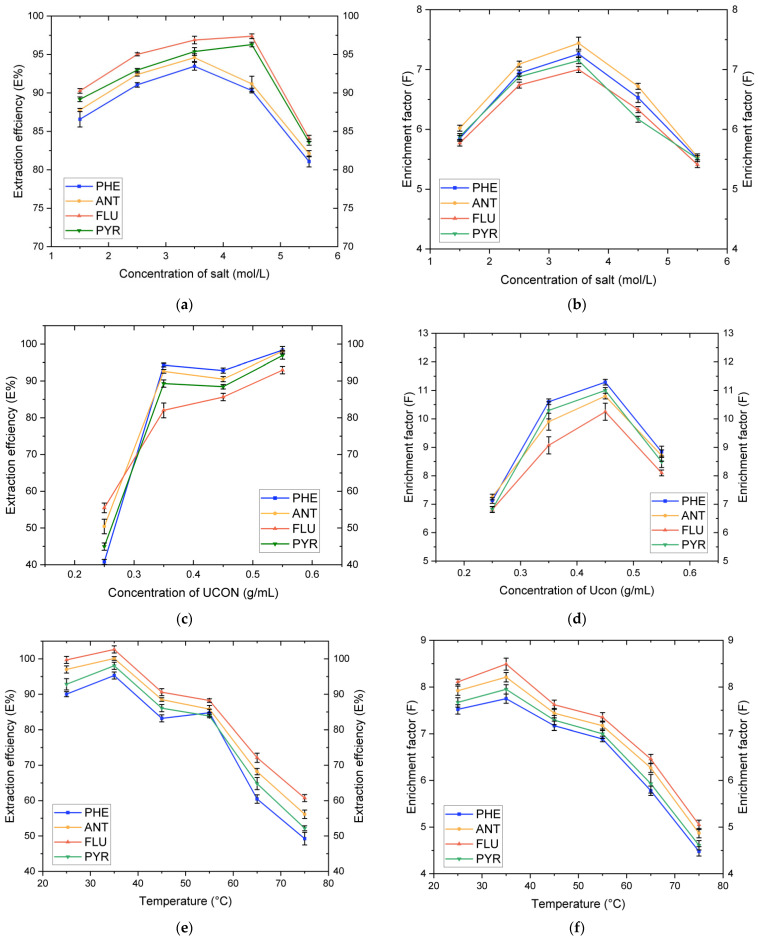
Effects of factors on extraction efficiencies and enrichment factors of FLU, ANT, PHE, and PYR: (**a**) Influence of the NaH_2_PO_4_ concentration on extraction efficiencies; (**b**) Influence of the NaH_2_PO_4_ concentration on enrichment factors; (**c**) Influence of the UCON concentration on extraction efficiencies; (**d**) Influence of the UCON concentration on enrichment factors; (**e**) Influence of temperature on extraction efficiencies; and (**f**) Influence of temperature on enrichment factors.

**Figure 4 molecules-27-06465-f004:**
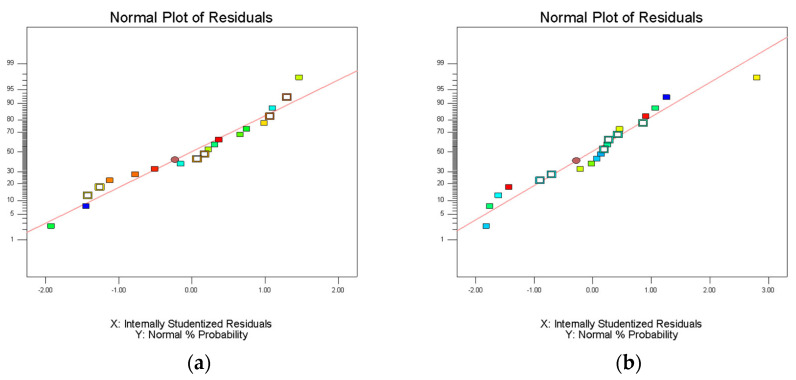
Normal plots of residuals of model fitting extraction efficiency and enrichment factor of PHE, ANT, FLU, and PYR: (**a**) extraction efficiency model of PHE; (**b**) enrichment factor model of PHE; (**c**) extraction efficiency model of ANT; (**d**) enrichment factor model of ANT; (**e**) extraction efficiency model of FLU; (**f**) enrichment factor model of FLU; (**g**) extraction efficiency model of PYR; and (**h**) enrichment factor model of PYR.

**Figure 5 molecules-27-06465-f005:**
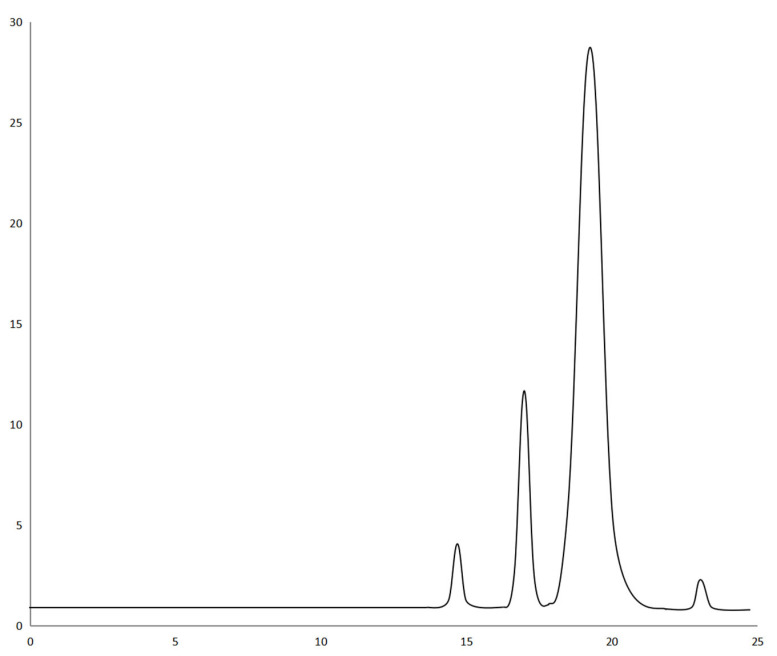
HPLC chromatogram with UV detection of PHE, ANT, FLU, and PYR.

**Figure 6 molecules-27-06465-f006:**
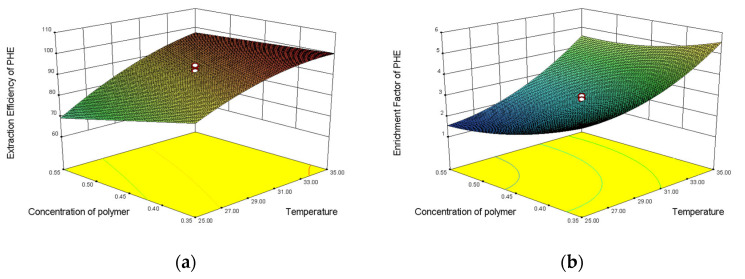
Response surface plots of extraction efficiency and enrichment factor of PHE, ANT, FLU, and PYR: (**a**) extraction efficiency model of PHE; (**b**) enrichment factor model of PHE; (**c**) extraction efficiency model of ANT; (**d**) enrichment factor model of ANT; (**e**) extraction efficiency model of FLU; (**f**) enrichment factor model of FLU; (**g**) extraction efficiency model of PYR; and (**h**) enrichment factor model of PYR.

**Figure 7 molecules-27-06465-f007:**
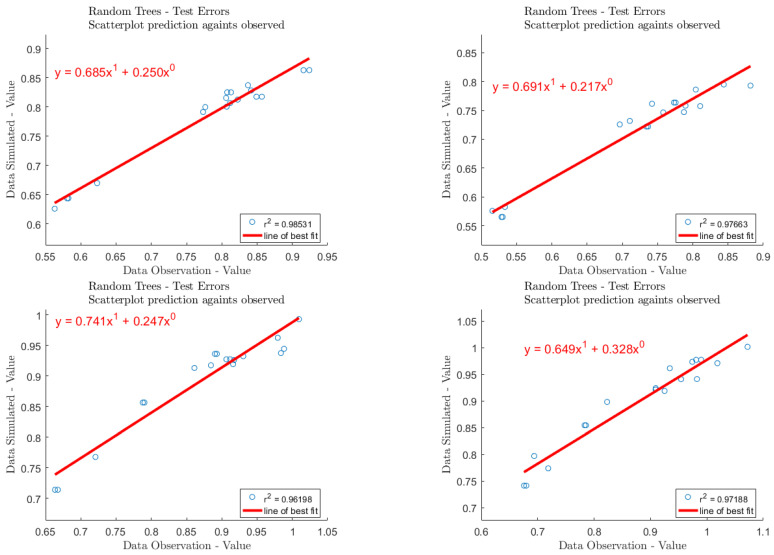
The regression R-values of the proposed ANN model used on the test dataset.

**Figure 8 molecules-27-06465-f008:**
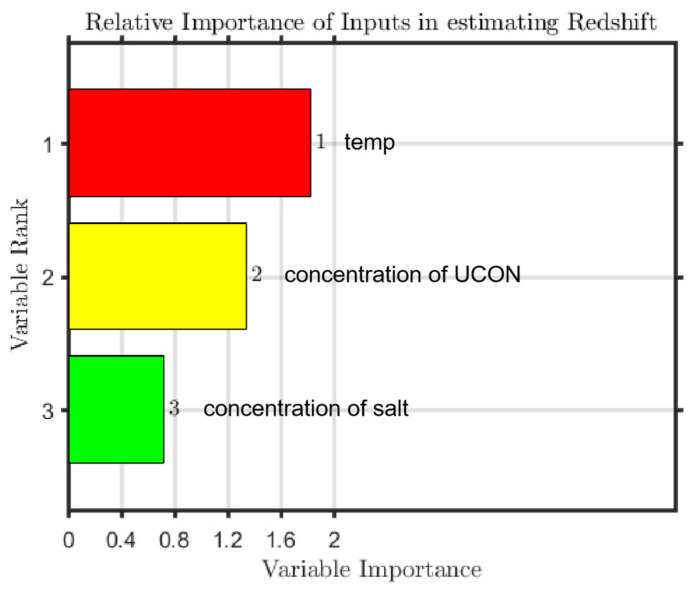
Sorted importance of input parameters.

**Figure 9 molecules-27-06465-f009:**
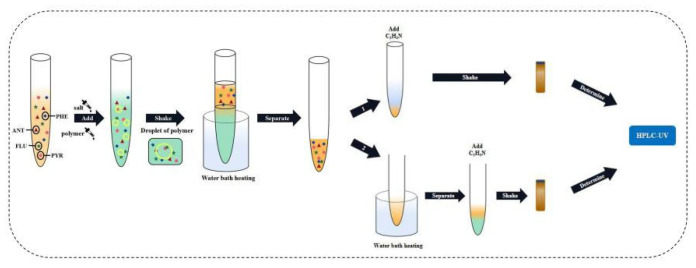
Flow chart showing the secondary separation and determination of four PAHs.

**Table 1 molecules-27-06465-t001:** Factors and their levels.

Level Factors			
	A Temperature (°C)	B Concentration of Polymer (g·mL^−1^)	C Concentration of NaH_2_PO_4_ (mol·L^−1^)
−1	25	0.35	2.5
0	30	0.45	3.5
1	35	0.55	4.5

**Table 2 molecules-27-06465-t002:** Orthogonal experiment scheme and experimental results.

RUN	X_1_	X_2_	X_3_	Y_11_	Y_21_	Y_31_	Y_41_	Y_12_	Y_22_	Y_32_	Y_42_
	(°C)	(g·mL^−1^)	(mol·L^−1^)	(E%)	(E%)	(E%)	(E%)	(F)	(F)	(F)	(F)
1	30	0.45	3.5	89.24 ± 0.16	91.13 ± 0.52	103.51 ± 0.32	91.19 ± 0.20	2.57 ± 0.03	2.20 ± 0.04	4.21 ± 0.07	2.81 ± 0.01
2	30	0.45	3.5	94.24 ± 0.34	88.64 ± 0.37	101.05 ± 0.44	86.81 ± 0.51	2.62 ± 0.05	2.46 ± 0.09	3.92 ± 0.06	2.30 ± 0.03
3	25	0.55	2.5	51.10 ± 0.29	52.26 ± 0.21	75.22 ± 0.52	58.25 ± 0.43	1.29 ± 0.07	3.92 ± 0.01	1.88 ± 1.04	2.00 ± 0.09
4	30	0.45	3.5	92.10 ± 0.51	91.57 ± 0.68	100.49 ± 0.15	90.6 ± 0.21	2.85 ± 0.09	2.05 ± 1.01	4.00 ± 0.04	2.66 ± 1.06
5	25	0.35	2.5	79.94 ± 0.94	77.72 ± 0.92	107.6 ± 0.71	75.06 ± 0.52	3.33 ± 0.03	3.05 ± 0.08	4.48 ± 0.05	3.2 ± 1.05
6	25	0.55	4.5	70.08 ± 0.37	60.81 ± 0.19	80.68 ± 0.63	68.16 ± 0.78	2.19 ± 0.02	1.50 ± 0.07	1.16 ± 1.04	1.43 ± 0.03
7	38.41	0.45	3.5	98.55 ± 0.59	95.68 ± 0.25	86.01 ± 0.22	95.55 ± 0.93	6.48 ± 1.01	3.99 ± 0.09	6.52 ± 0.02	5.20 ± 0.07
8	35	0.35	2.5	95.10 ± 0.28	83.95 ± 0.73	95.93 ± 0.21	84.8 ± 0.54	6.42 ± 0.03	5.50 ± 0.04	5.07 ± 0.08	4.74 ± 1.04
9	30	0.45	5.18	74.14 ± 0.66	66.82 ± 0.82	96.77 ± 0.30	76.63 ± 0.29	2.59 ± 0.09	2.40 ± 0.06	3.72 ± 0.03	3.19 ± 0.05
10	30	0.45	3.5	98.91 ± 0.81	99.72 ± 0.16	99.17 ± 0.11	99.84 ± 0.61	5.02 ± 0.04	6.23 ± 0.02	5.59 ± 0.07	4.60 ± 0.09
11	30	0.28	3.5	100.33 ± 0.53	99.18 ± 0.44	111.72 ± 0.24	91.98 ± 0.27	3.02 ± 0.05	1.7 ± 0.08	3.89 ± 1.05	2.86 ± 0.01
12	30	0.45	1.82	65.33 ± 0.23	63.89 ± 0.38	95.39 ± 0.62	71.62 ± 0.34	3.11 ± 0.01	3.04 ± 0.03	2.94 ± 1.01	2.98 ± 1.04
13	25	0.35	4.5	72.90 ± 0.64	70.68 ± 0.15	93.38 ± 0.45	73.6 ± 0.13	3.31 ± 1.05	3.21 ± 0.08	4.24 ± 0.02	3.35 ± 1.02
14	30	0.45	3.5	92.32 ± 0.79	89.35 ± 0.27	100.8 ± 0.17	92.73 ± 0.22	2.91 ± 1.01	2.1 ± 0.03	3.95 ± 0.06	2.2 ± 0.08
15	35	0.35	4.5	89.78 ± 0.77	85.39 ± 0.61	98.58 ± 0.25	86.84 ± 0.51	4.93 ± 1.01	7.1 ± 0.03	6.8 ± 0.09	5.4 ± 1.06
16	35	0.55	2.5	85.28 ± 0.38	88.89 ± 0.62	94.18 ± 0.15	81.8 ± 0.04	4.53 ± 0.14	3.2 ± 0.09	3.92 ± 0.11	3.4 ± 0.04
17	35	0.55	4.5	94.22 ± 0.16	90.58 ± 0.33	104.05 ± 0.13	94.45 ± 0.24	5.23 ± 1.06	1.7 ± 0.04	5.78 ± 0.10	3.5 ± 0.03
18	21.59	0.45	3.5	64.61 ±0.67	59.03 ± 0.47	73.07 ± 0.72	61.97 ± 0.63	2.31 ± 1.01	2.5 ± 0.08	2.61 ± 0.03	2.21 ± 0.14
19	30	0.62	3.5	84.4 ± 0.48	85.62 ± 0.26	95.67 ± 0.95	81.93 ± 0.38	2.34 ± 0.03	2.38 ± 0.09	2.66 ± 0.09	1.9 ± 0.04
20	30	0.45	3.5	88.89 ± 0,95	92.32 ± 0.32	101.23 ± 0.11	90.01 ± 0.61	2.87 ± 0.07	2.14 ± 0.02	3.95 ± 0.04	2.57 ± 1.01

Y_ki_ are the responses (k = 1 represents PHE, k = 2 represents ANT, k = 3 represents FLU, and k = 4 represents PYR; i = 1 represents extraction efficiency and i = 2 represents enrichment factor).

**Table 3 molecules-27-06465-t003:** Generalization ability of the proposed RF and RSM models.

ATPS	Object	Polymer or Alcohol	Salt	Temperature	Design of Experiment	Number of Data	MRPD		Ref.
							RF	RSM	
UCON –NaH_2_PO_4_	PHE	0.28~0.62 g·mL^−1^	1.82~5.18 mol·L^−1^	21.59~38.41 °C	CCD ^a^	20	0.646	8.632	Present study
	ANT						2.047	5.852	
	FLU						2.948	4.591	
	PYR						3.571	11.598	
Alcohol –NaH_2_PO_4_	flavor	28~36%	4.5~5.0 g·mL^−1^	25 °C	BBD ^b^	17	0.764	27.71	[38]
PEG-citrate –NaCl	a-amylase	9~19%	10~30%	room temperature	CCD	20	7.116	23.79	[39]
PEG-phosphate–NaCl	canavalia brasiliensis lectin	16.5~18.5%	17.5~21.5%	25 °C	CCD	26	4.378	25.06	[40]
POELE10 –(NH_4_)_2_SO_4_	thiamphenicol	0.021~0.033 g·mL^−1^	0.138~0.150 g·mL^−1^	288.15~308.15 K	CCD	26	7.832	15.13	[41]
POELE10 –NaH_2_PO_4_	chloramphenicol	0.021~0.033 g·mL^−1^	0.174~0.198 g·mL^−1^	15~35 °C	CCD	26	6.233	18.93	[31]
POELE10 –Na_2_C_4_H_4_O_6_	sulfadiazine	0.024~0.030 g·mL^−1^	0.162~0.180 g·mL^−1^	15~35 °C	OD ^c^	9	3.585	19.92	[32]
POELE10 –Na_2_C_4_H_4_O_6_	sulfamethazine	0.024~0.030 g·mL^−1^	0.162~0.180 g·mL^−1^	15~35 °C	OD	9	4.544	19.88	

^a^ CCD Central composite design. ^b^ BBD Box–Behnken design. ^c^ OD Orthogonal design.

**Table 4 molecules-27-06465-t004:** Recycling times and recovery of PAHs.

	Volume of UCON (mL)	Recovery of PHE	Recovery of ANT	Recovery of FLU	Recovery of PYR
First extraction	3.0	90.11 ± 0.04%	93.42 ± 0.18%	95.07 ± 0.22%	91.15 ± 0.17%
First UCON recycling	2.6	86.71 ± 0.13%	87.92 ± 0.28%	88.75 ± 0.25%	85.49 ± 0.07%
Second extraction	2.6 + 0.4	91.29 ± 0.26%	94.68 ± 0.13%	96.33 ± 0.61%	92.69 ± 0.52%
Second UCON recycling	2.6	86.93 ± 0.04%	88.04 ± 0.33%	89.13 ± 0.19%	85.96 ± 0.23%
Third extraction	2.6 + 0.4	91.77 ± 0.44%	95.20 ± 0.27%	96.82 ± 0.13%	92.93 ± 0.32%
Third UCON recycling	2.5	87.14 ± 0.61%	88.19 ± 0.35%	89.33 ± 0.14%	86.07 ± 0.30%

**Table 5 molecules-27-06465-t005:** Standard calibration data for the UCON ATPES–HPLC method for PHE, ANT, FLU, and PYR in spiked samples.

Analyte	Matrix	Linear Range	Linear Regression Equations	Relative Coefficient
Water				
Xiasantai River	PHE	0.1–100 ng mL^−1^	y = 47.155x + 15.1	0.9992
	ANT		y = 177.38x + 42.15	0.9999
	FLU		y = 462.67x + 85.218	0.9999
	PYR		y = 30.285x + 2.2	0.9997
Tashan Reservoir	PHE		y = 46.217x + 13.765	0.9992
	ANT		y = 175.61x + 41.209	0.9999
	FLU		y = 459.09x + 75.467	0.9998
	PYR		y = 29.982x + 1.658	0.9997
Soil				
Topsoil	PHE	0.1–100 ng g^−1^	y = 40.082x + 12.725	0.9992
	ANT		y = 153.3x + 14.673	0.9998
	FLU		y = 380.21x + 131.01	0.9992
	PYR		y = 25.758x + 1.1519	0.9996
Subsoil	PHE		y = 40.883x + 13.087	0.9992
	ANT		y = 156.36x + 15.073	0.9998
	FLU		y = 387.82x + 133.74	0.9992
	PYR		y = 26.273x + 1.2819	0.9996

**Table 6 molecules-27-06465-t006:** Comparison of different methods of calculation of extraction efficiency of four targets.

Method	Sample	Extraction Efficiency (%)				Recycling	References
		FLU	ANT	PHE	PYR		
GC-MS-SPE: graphene/chitosan composite aerogel	Trace polycyclic aromatic hydrocarbons in water and milk	97.2		96.5	95.5	No	[42]
Solid-phase extraction (SPE)	Polycyclic aromatic hydrocarbons in environmental aqueous matrices		80.9 ± 3.8		83.0 ± 13.4	No	[43]
Gas chromatography-triple quadrupole mass spectrometry with temperature-controlled ultrasonic extraction	Polycyclic aromatic hydrocarbons in atmospheric fine particulate matter (PM2.5)		90.7		90.1	No	[44]
GC-MS-dispersive solid phase extraction (d-SPE)	Natural coffee	62.2 ± 4.8	69.6 ± 10.3		78.5 ± 6.7	No	[45]
	Cereal coffee	76.3 ± 7.0	81.7 ± 10.1		85.9 ± 9.0		
	Smoked ham	54.2 ± 5.3	74.1 ± 8.4		65.0 ± 2.0		
	Smoked cheese	57.7 ± 2.4	60.3 ± 2.3		50.0 ± 4.0		
	Biscuits	85.1 ± 3.5	78.5 ± 3.9		71.3 ± 1.0		
	Crackers	86.9 ± 6.0	74.6 ± 1.3		72.4 ± 1.3		
ATPES: UCON–NaH2PO4-HPLC–UV	Water	98.075 ± 0.645	97.95 ± 1.24	97.9 ± 1.14	98.41 ± 0.89	Yes	This work
	Soil	98.36 ± 0.89	98.59 ± 1.25	98.48 ± 1.03	98.395 ± 0.835		

## Data Availability

Not applicable.

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
