# Peer review of "Simultaneous Prediction, Determination, and Extraction of Four Polycyclic Aromatic Hydrocarbons in the Environment Using a UCON–NaH2PO4 Aqueous Two-Phase Extraction System Combined with High-Performance Liquid Chromatography-Ultraviolet Detection"

_molecules, 2022, doi:10.3390/molecules27196465_

Round 1

Reviewer 1 Report

The manuscript describes the ATPE optimization to concentrate/isolate 4 PAHs = water contaminants (polycyclic aromatic hydrocarbons) for quantification in HPLC.

The development is described and illustrated using model solutions.

Validation is done using real samples from the "Xiasantai River".

The presentation of the results is good, comprehensively described and understandable.

However, the scope of the work is very narrow. Transferability is only briefly addressed in the last sentence of the Conclusion:

"The work also provides insights into whether this simultaneous extraction and separation method can be applied to other food or environmental samples."

The transferability and general relevance of the results as well as missing research work should be presented in an extended way.

It is urgently recommended to extend the reproducibility of the data as well as the results with regard to the error calculation as well as altogether all presentation of data by error bars.

I am sure the authors will quickly overcome these problems.

Author Response

My detailed responses to all of the points raised by the reviewers

Manuscript ID: molecules-1895797

Title:"Simultaneous prediction, determination, and extraction of four polycyclic aromatic hydrocarbons in the environment using a UCON-NaH2PO4 aqueous two-phase extraction system combined with high-performance liquid chromatography-ultraviolet detection"

Author(s): He Chang; Lu, Yang; Yantao Sun

Dear reviewers:

Thank you for your valuable suggestion. We have carefully revised our manuscript according to the reviewer’s comments. However, if you have any questions or are dissatisfied for whatever reason, I would greatly appreciate you giving me an opportunity to correct any wrongs.

Reviewer’s comments:

  1. The scope of the work is very narrow. Transferability is only briefly addressed in the last sentence of the Conclusion.The transferability and general relevance of the results as well as missing research work should be presented in an extended way.

Thanks for the reviewer’s suggestion. In this work, we only compared the generalization ability of the random forest and RSM two models constructed and the comparison of the extraction efficiency of the four targets under different extraction methods. There was no description about the content work of "The work also provides insights into whether this simultaneous extraction and separation method can be applied to other food or environmental samples.". So according to the reviewer’s suggestion, we have added the application of UCON-NaH2PO4 ATPES in the separation of trace sulfadiazine and sulfadimethazine from food and environment(3.5.2). The relevant experimental data have been showed in Supplementary Table 3. In addition, we will continue to perform other relevant validations to further clarify that this simultaneous extraction separation method can be applied to other food or environmental samples.(lines 564-575, page 21-22)

  1. It is urgently recommended to extend the reproducibility of the data as well as the results with regard to the error calculation as well as altogether all presentation of data by error bars.

According to the reviewer’s suggestion, we have given the error bars of the experiment data in figures and tables. (Figures 2-3, Tables 2 and 4)

Thank you again for your valuable suggestion and we are looking forward to your positive response!

Yours sincerely,

Yang Lu

Reviewer 2 Report

Dear authors,

Thank you very much for submitting your manuscript on "Simultaneous prediction, determination, and extraction of four polycyclic aromatic hydrocarbons in the environment using a UCON-NaH2PO4 aqueous two-phase extraction system combined with high-performance liquid chromatography-ultraviolet detection". 

The methods are described in very much detail, sometimes too many details.  

Some general remarks:

Page 4/27, line 171 - It is not "inverted phase" but "reversed phase", plese change.

Page 6/27, line 235 - "...composition of anion cation.."? Please check and correct. 

Page 7/27, line 278 - Which strength do you refer to?

Page 11/27 - The quadratic equations would be better supplied as Supportive Material, which I would also suggest for Tables 3 and 7.

The language of the manuscript needs, please contact a native English speaker.

At this time, I cannot recommend the publication of the manuscript.

Kund regards

Author Response

My detailed responses to all of the points raised by the reviewers

Manuscript ID: molecules-1895797

Title:"Simultaneous prediction, determination, and extraction of four polycyclic aromatic hydrocarbons in the environment using a UCON-NaH2PO4 aqueous two-phase extraction system combined with high-performance liquid chromatography-ultraviolet detection"

Author(s): He Chang; Lu, Yang; Yantao Sun

Dear reviewers:

Thank you for your valuable suggestion. We have carefully revised our manuscript according to the reviewer’s comments. However, if you have any questions or are dissatisfied for whatever reason, I would greatly appreciate you giving me an opportunity to correct any wrongs.

Reviewer’s comments:

  1. Page 4/27, line 171 - It is not "inverted phase" but "reversed phase", plese change.

According to the reviewer’s suggestion, we have made a careful revision.(lines 179, page 4)

  1. Page 6/27, line 235 - "...composition of anion cation.."? Please check and correct.

According to the reviewer’s suggestion, we have made a careful revision.(lines 244, page 6)

  1. Page 7/27, line 278 - Which strength do you refer to?

The ‘strength’ means ‘the aqueous phase is more stratified’. We have replaced ‘strength’ with ‘the aqueous phase is more stratified’ in this paper.(lines 288-289, page 7)

  1. Page 11/27 - The quadratic equations would be better supplied as Supportive Material, which I would also suggest for Tables 3 and 7.

According to the reviewer’s suggestion, the original tables 3 and 7 were provided in supplementary material(Supplementary Tables 1 and 2). And the quadratic equations were also provided in supplementary material.

  1. The language of the manuscript needs, please contact a native English speaker.

Thanks for the reviewer’s suggestion. This article has been commissioned by Zhengzhou Yuwen Education Consulting Co., Ltd. of China to do language polishing. However, we will use the ‘Language Editing Services’ recommended by MDPI to polish the manuscript if required.

Thank you again for your valuable suggestion and we are looking forward to your positive response!

Yours sincerely,

Yang Lu

Round 2

Reviewer 1 Report

The changes made by the authors are sufficient. I still would like to see an extended description about what the precise follow-up research needs to solve or what other challenges remain that need to be tackled in the conclusion. However, I recommend accept in present form.

Reviewer 2 Report

Dear authors,

Thank you for considering the suggestions and editing the manuscript.

Please, consider again a through check of the English language in the manuscript. 

Kind regards